# Prevalence of Physical Health, Mental Health, and Disability Comorbidities among Women Living with HIV in Canada

**DOI:** 10.3390/jpm12081294

**Published:** 2022-08-06

**Authors:** Emily Heer, Angela Kaida, Nadia O’Brien, Bluma Kleiner, Alie Pierre, Danielle Rouleau, Ann N. Burchell, Lashanda Skerritt, Karène Proulx-Boucher, Valerie Nicholson, Mona Loutfy, Alexandra de Pokomandy

**Affiliations:** 1Department of Public Health, McGill University, Montreal, QC H3A 1G1, Canada; 2Faculty of Health Sciences, Simon Fraser University, Vancouver, BC V5A 1S6, Canada; 3Department of Family Medicine, McGill University, Montreal, QC H3S 1Z1, Canada; 4Research Institute of McGill University Health Centre, Montreal, QC H4A 3S9, Canada; 5Department of Infectious Diseases, Centre Hospitalier de l’Université de Montréal, Montreal, QC H2X 0C1, Canada; 6Department of Family and Community Medicine, St Michael’s Hospital, Unity Health, Toronto, ON M5B 1W8, Canada; 7Women’s College Research Institute, Women’s College Hospital, Toronto, ON M5S 1B2, Canada; 8Faculty of Medicine, University of Toronto, Toronto, ON M5S 1A1, Canada; 9Chronic Viral Illness Service, McGill University Health Centre, Montreal, QC H4A 3J1, Canada

**Keywords:** HIV, women, comorbidity, disability, mental health, prevalence, ethnicity, sex, CHIWOS

## Abstract

Life expectancy for people living with HIV has increased, but management of HIV is now more complex due to comorbidities. This study aimed to measure the prevalence of comorbidities among women living with HIV in Canada. We conducted a cross-sectional analysis using data from the 18-months survey (2014–2016) of the Canadian HIV Women’s Sexual and Reproductive Health Cohort Study (CHIWOS). Self-report of diagnosed conditions was used to measure lifetime prevalence of chronic physical conditions, current mental health conditions, and disabilities. We examined frequency of overlapping conditions and prevalence stratified by gender identity, ethnicity, and age. Among 1039 participants, 70.1% reported a physical health diagnosis, 57.4% reported a current mental health diagnosis, 19.9% reported a disability, and 47.1% reported both physical and mental health comorbidities. The most prevalent comorbidities were depression (32.3%), anxiety (29.5%), obesity (26.7%, defined as body mass index >30 kg/m^2^), asthma/chronic obstructive pulmonary disease (23.3%), sleep disorder (22.0%), drug addiction (21.9%), and arthritis/osteoarthritis (20.9%). These results highlight the complexity of HIV care and the important prevalence of comorbidities. Personalized health care that integrates care and prevention of all comorbidities with HIV, with attention to social determinants of health, is necessary to optimize health and well-being of women living with HIV.

## 1. Introduction

The introduction of antiretroviral therapy (ART) has reduced complications associated with AIDS-defining illness and has significantly increased the life expectancy of people living with HIV [1]. As life expectancy has increased, the likelihood of acquiring additional chronic diseases has also increased [2]. Previous studies have demonstrated a higher risk of cardiopulmonary conditions, liver and renal disease, certain cancers, osteoporosis, and mental illness in people living with HIV compared to the general population [3,4]. In a Canadian study, one-third of the people with HIV had at least one other chronic condition, and more than one-third had a mental health condition diagnosed in the previous two years [5]. Further, more than half of deaths and clinical events among those on ART are classified as non-AIDS related [6].

The mechanisms resulting in high prevalence of comorbidity in people with HIV are multifactorial and may be linked to the HIV virus itself, long-term toxicity of ART, immune activation, chronic inflammation, and the intersection of unfavorable social determinants of health [6,7]. For example, lung diseases such as chronic obstructive pulmonary disease (COPD) and lung cancer occur at higher rates among those with HIV. The increased incidence is not only due to the higher prevalence of cigarette smoking in this population, but it has also been linked to impaired immune function and chronic inflammation related to the virus, often termed inflammaging [8,9,10,11]. Additionally, there is evidence that people with HIV undergo an accelerated aging process whereby conditions traditionally associated with aging (e.g., heart diseases, low bone mineral density) progress at a faster rate [2]. In addition to HIV care, primary care of people living with HIV must integrate prevention and management of comorbidities; recognizing these needs is essential to plan healthcare services and multidisciplinary resources [12].

Within a global context, traditional gender roles may also impact treatment and adherence to ART [13,14,15]. Women are often caregivers for children, elderly parents, and extended family members and may not prioritize their own medical care; as a result, they may be more vulnerable to developing serious health problems [16]. Women may face unequal power, including gender-based economic decision-making in relationships, which can lead to isolation, depression, and a reduced ability to adhere to ART and seek routine medical care, adding to the intersectionality with other unfavorable social determinants of health [15,17]. In Canada, women constitute nearly a quarter of reported HIV cases [18]. However, few women are included in studies investigating comorbidities among people living with HIV, precluding sex- and gender-specific analyses of health outcomes [19]. A previous study conducted in Ontario showed that women had higher rates of comorbidity across all age groups as compared with women not living with HIV [5]. Other studies found that females were more susceptible to developing low bone density and osteoporosis, and these may be further exacerbated by HIV and ART [4,20]. A systematic review recently highlighted the paucity of data regarding women living with HIV, particularly above 50 years old [21]. Age, ethnicity, gender, and biological sex are all factors that must be considered when providing person-centered care for patients, including those living with HIV.

The objective of this study was to measure the prevalence of comorbidities among women living with HIV in Canada overall, stratified by gender identity, and by age and ethnicity.

## 2. Materials and Methods

We conducted a cross-sectional analysis using data from the 18-months survey (2014–2016) of the Canadian HIV Women’s Sexual and Reproductive Health Cohort Study (CHIWOS). CHIWOS is a multi-site, longitudinal, community-based research study conducted across Canada (www.chiwos.ca, last accessed on 5 July 2022) [22]. Persons self-identifying as women living with HIV (cis, trans, intersex, two-spirit, gender queer), ≥16 years of age, and living in British Columbia (BC), Ontario, and Quebec were recruited through peer word-of-mouth, HIV clinics, AIDS Service Organizations, non-HIV community-based organizations (immigration and refugee services, women’s shelters, harm reduction and sex worker support services), and online methods such as social network sites [23]. Data were collected using a structured online questionnaire administered by peer research associates; these were women living with HIV integrated into the research study team. Peer research associates were trained in research ethics and survey interview techniques [22]. Surveys were conducted in person, in French or English, at a confidential setting or over Skype or telephone [24]. All participants provided voluntary informed consent at baseline. Primary ethics approval was provided by the Research Ethics Boards of Simon Fraser University, University of British Columbia/Providence Health Care, Women’s College Hospital, McGill University Health Centre. Other recruiting study sites also obtained their own approval prior to enrolment.

Participants self-reported their gender identity, and sex at birth. Although all participants in CHIWOS identified as women, for the purposes of this article, gender was dichotomized as transgender women (self-identified as transwoman and reported male, intersex sex or unknown sex at birth) and cisgender women (indicated sex assigned at birth was female or undetermined and self-identified as a woman, two-spirited, queer or other without further specifying).

Prevalence of a chronic physical condition, a current mental health condition, or a current disability was determined via self-reported responses to the 18-month surveys. We used the 18-month survey data, rather than baseline, as it contained the most comprehensive questions regarding comorbidities. Prevalence of physical health conditions was measured for multiple chronic diseases. We asked: “Have you ever been diagnosed with any of the following health concerns?” The participant could then select one or more of the following responses: asthma/Chronic Obstructive Pulmonary Disease (COPD), thyroid problems, coronary artery disease, cardiac arrhythmia, osteoporosis/osteopenia/decreased bone density, fractures, stroke, deep vein thrombosis and pulmonary embolism, high cholesterol, hypertension, irritable bowel disease, renal problems, arthritis and osteoarthritis, chronic pain, and cognitive impairment. Self-reported height and weight was also used to identify obesity, defined as a body mass index (BMI) of >30 kg/m^2^. Although we recognize that some of these conditions can present as more acute than chronic (e.g., cardiac arrhythmia and deep vein thrombosis), these conditions often require lifelong attention in healthcare; therefore, they were included in the questionnaire.

For previous history of cancers, we asked: “Have you ever been diagnosed with any form of cancer or pre-cancer?” If a participant answered yes, they could select one of more of the following responses: oral or pharynx, thyroid, skin, lung, breast, liver, stomach or small bowel, colon or rectum, anal, ovarian, endometrial (i.e., of the uterus), cervical, vulvar, lymphoma/leukemia, bladder, kidney, high-grade cervical pre-cancer (Cervical Intraepithelial Neoplasia or CIN 2 or 3), high-grade vulvar or vaginal pre-cancer (Vulvar or Vaginal Intraepithelial Neoplasia, VIN or VaIN 2 or 3), high-grade anal pre-cancer (Anal Intraepithelial Neoplasia, AIN 2 or 3), and other (please specify). Note that since the time period we designed the questionnaire, the preferred terminology to designate high-grade precancer changed to high-grade squamous intraepithelial lesion (HSIL) [25], which is the terminology we will use from here on out.

For mental health conditions, we asked: “Which, if any, of the following mental health conditions are you currently living with? Please only include conditions that have been diagnosed by a health care provider”. Participants could select one or more of the following options: alcohol addiction, anxiety, anorexia or bulimia nervosa, bipolar disorder, personality disorder, dementia, depression, drug addiction, obsessive compulsive disorder, post-traumatic stress disorder (PTSD), schizophrenia, and sleep disorders.

For disabilities, we asked: “Do you have any of the following disabilities?”. Participants could select one or more of the response options: partial deafness, complete deafness, partial blindness, complete blindness, physical difficulty to walk (requiring assistive device like cane or walker on a regular basis), physical difficulty to walk (requiring wheelchair), speech difficulty, physical difficulty moving one or both arms, other (specify), don’t know, or prefer not to answer.

Descriptive statistics were performed to present sociodemographic characteristics of the study participants including age (16–29, 30–39, 40–49, ≥50), ethnicity (white, Indigenous, African, Caribbean, or Black (ACB), or other), country of birth (Canadian born, Foreign born), gender identity (cisgender or transgender), education level (<high school or ≥high school), employment (yes or no), household annual income (<20,000 CAD or ≥20,000 CAD), cigarette smoking status (never, current, or former smoker), injection drug use (ever or never), ever incarcerated (yes or no), and HIV viral load (undetectable (<50 copies/mL) or detectable (≥50 copies/mL)). We presented the characteristics of participants who completed the 18-months survey as well as of those who were lost to follow-up to acknowledge potential selection bias. All variables were self-reported. We also used descriptive statistics to measure the prevalence of comorbidities (self-reported physical and mental health conditions and disabilities) in this population, stratified by gender identity (cisgender vs. transgender) and by age and ethnicity. All proportions are presented with 95% confidence intervals (CI). Median age is presented with Interquartile Ranges (IQR). Proportions were considered to differ with statistical significance when the 95% CI did not overlap. Bar graphs and pie charts were also used to present results visually. All statistical analyses were done using Stata software version 15.1 (StataCorp LLC, College Station, TX, USA).

## 3. Results

### 3.1. Characteristics of Participants

We included a total of 1039 participants who completed the 18-month follow-up survey (Table 1). This represents 85% of all participants who completed the baseline survey in the CHIWOS study. The characteristics of participants retained versus those lost to follow-up are presented in Table 1; those lost to follow-up group had a higher proportion of participants living in Ontario, of younger age, of Indigenous ethnicity, born in Canada, who injected drugs, and who currently smoke. The median age of the retained participants was 45 years (IQR: 37–51); 39.8% identified as white, 16.7% as Indigenous, and 36.1% as ACB; 96.3% identified as cisgender women, and the majority (61.7%) had an annual household income of less than CAD 20,000. When examining characteristics of transgender participants specifically, the median age was 44.5 (IQR: 37–48); 28.1% identified as white, 25.0% as Indigenous, 15.6% as ACB, 31.3% as other ethnicities; and 87.5% reported an annual household income of less than CAD 20,000.

### 3.2. Lifetime Prevalence of Physical Health Diagnoses Overall and by Gender Identity

Table 2 presents the prevalence of diagnosed lifetime physical health conditions, mental health diagnoses currently living with, and disabilities currently living within the cohort of 1039 participants and stratified by gender identity. Our data identified that 70.1% (*n* = 728, 95% CI 67.2–72.8) of participants had at least one diagnosed physical health condition other than HIV infection. The most prevalent lifetime physical diagnoses were obesity (BMI > 30 kg/m^2^) in 26.7%, followed by asthma/COPD in 23.3%, arthritis/osteoarthritis in 20.9%, cancer or pre-cancer in 19.9%, chronic pains for other causes than arthritis/osteoarthritis and requiring medication in 19.1%, hypertension in 17.2%, high cholesterol in 13.4%, osteoporosis or osteopenia in 11.6% (9.4% reported previous fractures), thyroid problems in 10.3%, diabetes in 8.7%, HIV/AIDS wasting syndrome in 6.8%, and 6.2% reported a previous diagnosis of cardiac arrhythmias. The other physical conditions were selected by less than 5% (deep vein thrombosis or pulmonary embolism, renal problems, coronary artery diseases, inflammatory bowel disease, cognitive impairment, strokes, and fibromyalgia). Transgender women participants were a little younger (25.0% were 16–39, 43.8% were 40–49 and 31.3% were 50 or more), but the difference with cisgender participants was not statistically significant. However, the groups were statistically different in terms of ethnicity with a higher proportion of transgender women identifying as Indigenous (25.0% vs. 16.4%) or other (31.3% vs. 6.7%, Latin American being the most common), but a lower proportion identified as ACB (15.6% vs. 36.7%) or white (28.1% vs. 40.2%). There was no statistically significant difference in prevalence reported by transgender women participants as compared with cisgender women participants, although a lower proportion of transgender women reported arthritis/osteoarthritis (12.5% vs. 21.2%), cancer or pre-cancer (9.4% vs. 20.3%), hypertension (6.3% vs. 17.6%), and a higher proportion reported cardiac arrythmia (15.6% vs. 5.9%) or strokes (9.4% vs. 3.6%).

### 3.3. Prevalence of Mental Health Diagnoses Currently Living with Overall and by Gender Identity

At least one current mental health diagnosis was reported by 57.4% (*n* = 596, 95% CI 54.3–60.4) of participants. As presented in the second section of Table 2, the most common mental health diagnoses were depression in 32.3%, anxiety in 29.5%, sleep disorder in 22.0%, drug addiction in 21.9%, PTSD in 13.9%, alcohol addiction in 9.1%, and bipolar disorder in 6.4%. Other diagnoses (personality disorder, obsessive compulsive disorder, anorexia or bulimia nervosa, schizophrenia, and dementia) were reported by less than 5% of participants. Of note, psychosis and attention deficit hyperactivity disorder (ADHD) were also included in other specified responses by <5%. Transgender women participants reported a lower prevalence of sleep disorder than cisgender women participants (6.3% vs. 22.5%, close to statistical significance).

### 3.4. Disabilities Overall and by Gender Identity

Overall, 19.9% (*n* = 207, 95% CI 17.5–22.5) of participants reported living with a physical disability. Partial deafness was reported by 8.8%, and difficulty walking and requiring regular use of an assistive device such as a cane or walker was reported by 7.8% (Table 2). Other disabilities were reported by less than 5% (complete deafness, partial blindness, difficulty walking—require wheelchair, speech difficulty, difficulty moving one or both arms), and no statistically significant difference was observed between gender identity groups.

### 3.5. Cancers or Precancers Overall

Of the 1039 participants, 19.9% reported a previous cancer or precancer diagnosis; the specific types are presented in Figure 1.

The most common type reported was cervical cancer, reported by 10.0% (*n* = 104), but only 24 reported cervical high-grade squamous intraepithelial lesions (HSIL). Because human papillomavirus (HPV) related cancer and pre-cancer (defined as HSIL) can be difficult to distinguish for participants, they are reported together to reduce the risk of misclassification. A previous diagnosis of cervical cancer or HSIL was reported by 10.0% (*n* = 104, 95% CI 8.3–12.0), anal cancer or HSIL was reported by 0.8% (*n* = 8, 95% CI 0.3–1.5), and vulvar or vaginal cancer or HSIL was reported by 0.2% (*n* = 2, 95% CI 0–0.7). For other cancers, the most commonly reported were skin (1.4%, *n* = 14, 95% CI 0.7–2.3), endometrial (1.3%, *n* = 13, 95% CI 0.6–2.1), breast (1.3%, *n* = 13), ovarian (1.2%, *n* = 12, 95% CI 0.6–2.0), and lymphoma or leukemia (1.1%, *n* = 11, 95% CI 0.5–1.9), bone (0.5%, *n* = 5, 95% CI), colon or rectum (0.5%, *n* = 5, 95% CI), oral or pharynx (0.4%, *n* = 4, 95% CI), thyroid, and lung. Other cancers were reported by less than 0.2% of participants (liver, stomach or small bowel, Kaposi sarcoma, bladder, kidney, brain).

### 3.6. Physical, Mental Health Conditions and Disabilities Stratified by Ethnicity and Age

To further inform care for women living with HIV, we stratified comorbidities by ethnicity (Indigenous, ACB, white/other) and age categories (Table 3). Only statistically significant results are presented here unless presented as appearing different. The prevalence of most physical conditions and disabilities increased with age, while that of mental health conditions remained relatively stable across ages within an ethnic group. In contrast, obesity, appeared more prevalent in younger groups for women identifying as Indigenous, and white/other. However, the prevalence of obesity increased with age in ACB women, resulting in a higher prevalence in women aged 50 or more compared to the other ethnic groups (42.7% in ACB versus 20.3% in Indigenous and 15.9% in white/other). Asthma/COPD, arthritis/osteoarthritis, and chronic pains of other causes are more prevalent in Indigenous women, and white/other women aged 40–49 and 50 or more compared to ACB women of the same age. Cancer or precancer was reported more frequently by white/other women aged 16–39 and 40–49 compared to ACB women of the same age (25.3% and 25.8% vs. 8.7 and 9.9%). High cholesterol, and osteoporosis/osteopenia were also reported more frequently by white/other women compared to ACB women in the group aged 50 or more (25.8% vs. 12.8% for cholesterol, 23.6% vs. 11.1% for osteoporosis/osteopenia). Depression and anxiety were at least twice as prevalent in Indigenous and white/other women compared to ACB women across all age categories. Sleep disorders were also more common in white/other women compared to ACB women across all age groups, and the same difference was also observed between Indigenous and ACB women for the 40–49 age group. Indigenous women had the highest prevalence of drug addiction across all age groups (up to 67.3% in youngest age group, compared to 35.4% in white/other women, and <5% in ACB women), and the same was observed for alcohol addiction in women aged 40–49 and 50 or more. PTSD was also more prevalent in the Indigenous group aged 50 or more (30.5%) compared to white/other women (12.5%) and ACB women of the same age (5.1%). No statistically significant differences in prevalence of disabilities were observed across age and ethnicity.

### 3.7. Overall Burden of Overlapping Health Conditions

Figure 2 shows the proportion of women experiencing overlapping physical, mental health and/or disability conditions likely to require healthcare attention. Amongst women in the cohort, 32.0% (*n* = 332, 95% CI 29.1–34.9) reported both mental and physical health conditions (without disability); 19.5% (*n* = 203, 95% CI 17.2–22.1) reported a concurrent physical health condition (without disability or mental health condition); 15.1% (*n* = 157, 95% CI 13.0–17.4) reported a concurrent physical, mental health and disability conditions; 9.3% (*n* = 97, 95% CI 7.6–11.3) reported a concurrent mental health condition (without disability or physical health condition); 3.5% (*n* = 36, 95% CI 2.4–4.8) reported both physical health and disability conditions (without mental health condition); and 1.3% (*n* = 14, 95% CI 0.7–2.3) reported a disability with or without a concurrent mental health condition (but no concurrent physical health condition). Finally, only 19.2% (*n* = 200) reported the absence of any comorbidity in addition to HIV.

## 4. Discussion

Among Canadian women living with HIV, there is a significant burden of concurrent comorbidities in addition to HIV, with 70% currently living with at least one concurrent physical health condition, 20% living with a disability, and 57% living with at least one mental health condition. As a result, 81% of women living with HIV require health care attention for at least one condition other than HIV. Most physical health and disability conditions increased with age, but the prevalence of mental health conditions remains more stable across age groups. Obesity is the most prevalent physical comorbidity, and while ACB women aged 50 or more had the highest prevalence, younger groups had the highest prevalence amongst Indigenous and white/other women. Depression and anxiety are also very common, as they currently affect a third of our study population. The burden of many conditions varies widely across ethnicities and should be considered when caring for women living with HIV. Our data confirms that age, ethnicity, and gender identity are associated with differences in prevalence of comorbidities among a cohort of women living with HIV. These should not be seen as causal factors but should be used by clinician as guides to increase their index of suspicion when treating patients of these gender, age, and ethnicities. Our findings further support that all of a person’s life context, including their social determinants of health, must be considered when caring for women living with HIV. HIV primary care must also include integrated/coordinated care for the prevention and management of all comorbidities that may further impact health and wellbeing.

The high prevalence of comorbidities among women living with HIV in our cohort aligns with results of other studies, most conducted in United States [21,26]. Compared to the general Canadian population, the prevalence among women living with HIV in our study was similar for asthma/COPD, arthritis/osteoarthritis, hypertension, dyslipidemia, osteoporosis, and diabetes [27,28,29,30,31,32]. The lifetime prevalence of cancer among Canadians aged 12 or more reported by the public health agency of Canada is 7.1%. It is difficult to conclude whether our reported lifetime prevalence is higher despite the 10.0% prevalence of cervical cancer alone, because it is possible that misclassification occurred due to the difficulty for participants to differentiate between cancer and precancer and the likelihood that HSIL may have been reported as cancer. Our reported prevalence reinforces the importance of cervical cancer screening in this population; screening is a preventive standard of care that is not optimally achieved as we previously reported [33]. The prevalence of mental health conditions reported in our population was higher than those of the general Canadian population. While the 12-month prevalence of depression in the Canadian population, regardless of gender, is 4.7%, and while the lifetime prevalence is 11.2% [34], 32% of the women in our population reported currently living with depression. Similarly, the 12-month prevalence of anxiety in the Canadian population is 2.5% [35], while 30% of our population reported currently living with anxiety. For PTSD, the 1-month prevalence rate in the Canadian population is 2.4% [36], while almost 14% of our population reported currently living with it. Furthermore, the CHIWOS team previously measured a PTSD prevalence of 47.1% in this study population, using more specific PTSD scales in the questionnaires [37,38]; this demonstrates the important underreporting and underdiagnosis of PTSD among women living with HIV. We did not measure statistically significant differences of comorbidity prevalence by gender identity in our study population. Considering that only 32 participants identified as transgender women, we are likely underpowered to detect such difference. Nevertheless, we considered it important to present this data stratified by gender identity due to previously reported comorbidities experienced by transgender women living with HIV [38,39]. Larger studies are required to appropriately compare the prevalence of comorbidities between cisgender and transgender women living with HIV. The differences in prevalence observed across ethnicities are most likely due to confounding factors that were not measured, and these likely intersect with other social determinants of health to heighten the prevalence of multiple comorbidities. While our study does not demonstrate causality, our results can serve as flags to clinicians that each patient has a unique risk of comorbidity, often based on life conditions and medical challenges, and that our index of suspicion may be adapted to the person’s socio-demographic profile. HIV care providers must ensure the treatment of other conditions in addition to HIV, often in coordinated collaboration with another healthcare provider [12].

The results of this study have important preventative and primary care implications. The development of several chronic health conditions has been linked to decreased quality of life and a shorter life expectancy [40]. The resulting polypharmacy may also increase adverse effects related to drug–drug interactions [41,42] and decrease adherence to ART [43]. Interventions focused on prevention of and screening for comorbidities, alongside discussion of non-pharmaceutical treatment options, must be optimally used and integrated into routine HIV care to better serve this population [44,45,46,47,48]. Our results highlight the need to offer women living with HIV comprehensive person-centered care, either by integrating prevention and management of comorbidities as part of routine HIV care, or by facilitating access to care through coordinated care across providers and care sites [21,49]. Given the evidence that women with HIV experience the highest rates of depression and depressive symptoms among all categories of age and gender, and given the negative impact of mental health on the maintenance of physical health [40,45], mental health care must be integrated into or facilitated through HIV care [44]. It is also important to recognize that women living with HIV often experience intersectionality of unfavorable social determinants of health; this may further increase the risk of disease and decrease the likelihood of accessing medical care [50,51]. Therefore, social support must be part of integrated HIV care to help minimize the impact of unfavorable social determinants of health [52].

There are several limitations to this study. Self-reporting of diagnosed conditions may lead to misclassification in different directions. Cervical cancer may be overreported due to the confusing precancer terminology. Osteopenia might be underreported if discussions with the physicians were understood as an absence of osteoporosis. Absence of diagnosis may be due to lower interaction with medical care, but it does not always indicate the absence of conditions. Conditions may be present but undiagnosed, leading to a lower estimated prevalence. This may be the case for some ACB women who were not born in Canada and who have immigrated more recently. Lifetime report means these conditions may have been diagnosed before HIV diagnosis, meaning causality of HIV cannot be assessed. Social desirability bias might have decreased self-reporting of cigarette smoking, drug use, weight (used for BMI), and mental health conditions; however, the questionnaires were administered by peer research associates rather than health professionals which was expected to reduce social desirability bias. Still, the self-reported prevalence of mental health conditions remains concerning. Recall bias for mental health diagnoses was reduced by conducting analyses on conditions participants were currently living with and should not affect the reporting of chronic physical health condition or disabilities. The external generalizability of our results is limited to women already engaged in care, and even though CHIWOS recruited approximately 10% of all women living with HIV in Canada, women were not sampled randomly and may not be representative of all women living with HIV in Canada. Finally, although we drew from the 18-month survey whose responses pertaining to comorbidities were more complete, this survey is affected by selection bias of participants remaining in the study. Indeed, the participants lost to follow-up were younger but had more unfavorable determinants of health, making it possible that our results underestimate the burden of comorbidities in the entire population of women living with HIV in Canada. Finally, our sample of transgender women was very small and out study is likely underpowered to detect differences in comorbidity prevalence.

## 5. Conclusions

Our results highlight the need for personalized HIV care that integrates the prevention and management of comorbidities. Considering women living with HIV are living longer due to more effective therapies, personalized HIV care must consider the dynamic nature of social determinants of health across the lifespan. Person-centered and stigma-free approaches are essential to consider the different risk factors for different health conditions experienced by women living with HIV. Clinicians’ index of suspicion should also be adapted to the socio-demographic profile of patients to ensure conditions are not missed and the care meets the needs of each person. Obesity, depression, and anxiety are particularly concerning, each affecting a quarter to a third of women participating in the CHIWOS study, but close to half for certain specific age and ethnic groups. These findings also justify the need for multidisciplinary health care teams to optimize the health and wellbeing of women living with HIV.

## Figures and Tables

**Figure 1 jpm-12-01294-f001:**
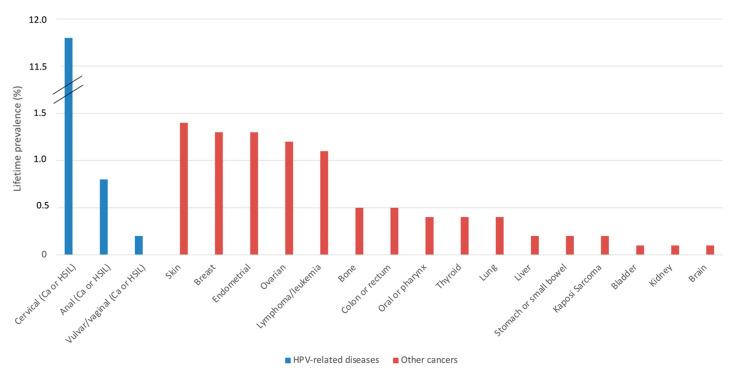
Prevalence of self-reported lifetime diagnoses of cancers or pre-cancers in participants from the Canadian HIV Women’s Sexual and Reproductive Health Cohort Study (CHIWOS, *n* = 1039).

**Figure 2 jpm-12-01294-f002:**
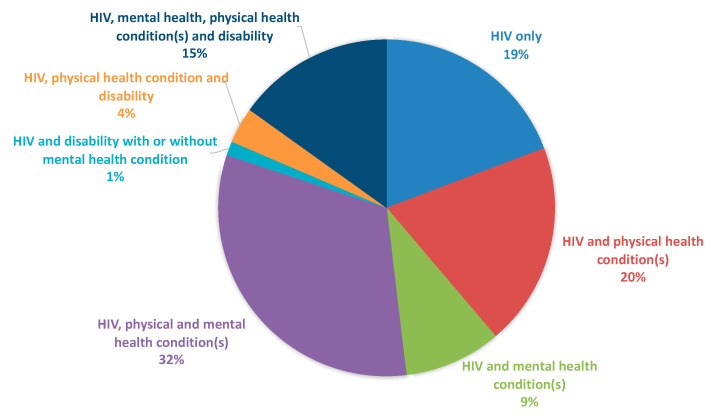
Overlapping burden of physical, mental health and disability conditions in participants from the Canadian HIV Women’s Sexual and Reproductive Health Cohort Study (CHIWOS, *n* = 1039).

**Table 1 jpm-12-01294-t001:** Baseline characteristics of participants from the Canadian HIV Women’s Sexual and Reproductive Health Cohort Study (CHIWOS) who completed the 18-months survey compared to those who were lost to follow-up.

Characteristics	18-Months Survey Completed *n* (%) [95% CI]	Lost to Follow-Up *n* (%) [95% CI]
TOTAL	1039 (85.4)	178 (14.6)
Province		
British Columbia	299 (28.8) [26.0–31.6]	57 (32.0) [25.2–39.4]
Ontario	420 (40.4) [37.4–43.5]	88 (49.4) [41.9–57.0]
Québec	320 (30.8) [28.0–33.7]	33 (18.5) [13.1–25.0]
Age		
16–29	71 (6.8) [5.4–8.5]	20 (11.2) [7.0–16.8]
30–39	272 (26.2) [23.5–29.0]	53 (39.8) [23.2–37.1]
40–49	355 (34.2) [31.3–37.1]	66 (37.1) [30.0–44.6]
50+	341 (32.8) [30.0–35.8]	39 (21.9) [16.1–28.7]
Ethnicity		
Indigenous	173 (16.7) [14.4–19.1]	51 (28.7) [22.1–35.9]
ACB	375 (36.1) [33.2–39.1]	39 (21.9) [16.1–28.7]
White	414 (39.8) [36.9–42.9]	70 (39.3) [32.1–46.9]
Other	77 (7.4) [5.9–9.2]	18 (10.1) [6.1–15.5]
Country of birth		
Canadian-born	589 (56.8) [53.6–59.7]	124 (69.7) [62.3–76.3]
Foreign-born	448 (43.1) [40.1–46.2]	54 (30.3) [23.7–37.7]
DK/PNTA	2 (0.2) [0–0.7]	0 (0) [0–2.1]
Gender identity		
Cisgender woman	1001 (96.3) [95.0–97.4]	171 (96.1) [92.1–98.4]
Transgender woman	32 (3.1) [2.1–4.3]	5 (2.8) [0.9–6.4]
Other	6 (0.6) [0.2–1.3]	2 (1.1) [0.1–4.0]
Education		
Less than high school	181 (17.4) [15.2–19.9]	37 (20.8) [15.1–27.5]
High school or more	854 (82.2) [79.7–84.5]	140 (78.7) [71.9–84.4]
DK/PNTA	4 (0.4) [0.1–1.0]	1 (0.6) [0–3.1]
Employment		
Yes	233 (22.4) [19.9–25.1]	27 (15.2) [10.2–21.3]
No	800 (77.0) [74.3–79.5]	150 (84.2) [78.1–89.3]
DK/PNTA	6 (0.6) [0.2–1.3]	1 (0.6) [0–3.1]
Household income (CAD)		
<20,000	641 (61.7) [58.7–66.7]	121 (68.0) [60.6–74.8]
≥20,000	366 (35.2) [32.3–38.2]	47 (26.4) [20.1–33.5]
DK/PNTA	32 (3.1) [2.1–4.3]	10 (5.6) [2.7–10.1]
Injection drug use		
Never	687 (66.1) [63.2–69.0]	95 (53.4) [45.8–60.9]
Ever	348 (33.5) [30.6–36.5]	76 (42.7) [35.3–50.3]
DK/PNTA	4 (0.4) [0.1–1.0]	7 (3.9) [1.6–7.9]
Cigarette smoking status		
Never	428 (41.2) [38.2–44.3]	50 (28.1) [21.6–35.3]
Former	138 (13.3) [11.3–15.5]	21 (11.8) [7.5–17.5]
Regular/occasional smoker	469 (45.1) [42.1–48.2]	103 (57.9) [50.3–65.2]
DK/PNTA	4 (0.4) [0.1–1.0]	4 (2.3) [0.6–5.7]
Most recent HIV viral load		
Undetectable (<50 copies/mL)	861 (82.9) [80.4–85.1]	135 (75.8) [68.9–81.9]
Detectable (≥50 copies/mL) ^2^	134 (12.9) [10.9–15.1]	31 (17.4) [12.2–23.8]
DK/PNTA	44 (4.2) [3.1–5.6]	10 (5.6) [2.7–10.1]

Note: ACB = African, Caribbean, Black; DK/PNTA = Don’t know or prefer not to answer; CI = confidence intervals. ^2^ Includes those who never accessed HIV medical care

**Table 2 jpm-12-01294-t002:** Prevalence of comorbidities in participants from the Canadian HIV Women’s Sexual and Reproductive Health Cohort Study (CHIWOS) in decreasing order, and stratified by gender identity.

Diagnosed Health Condition ^1^	Overall	Gender Identity	
		Transgender Women	Cisgender Women
	*n* = 1039 *n* (%) [95% CI]	*n* = 32 *n* (%) [95% CI]	*n* = 1007 *n* (%) [95% CI]
**Physical health diagnosis (lifetime prevalence)**			
Obesity (BMI > 30)	277 (26.7) [24.0–29.5]	7 (21.9) [9.3–40.0]	270 (26.8) [24.1–29.7]
Asthma/COPD	242 (23.3) [20.8–26.0]	7 (21.9) [9.3–40.0]	235 (23.3) [20.8–26.1]
Arthritis, osteoarthritis	217 (20.9) [18.5–23.5]	4 (12.5) [3.5–29.0]	213 (21.2) [18.7–23.8]
Cancer or pre-cancer	207 (19.9) [17.5–22.5]	3 (9.4) [2.0–25.0]	204 (20.3) [17.8–22.9]
Chronic pains for other causes than arthritis/ osteoarthritis requiring medication	198 (19.1) [16.7–21.6]	4 (12.5) [3.5–29.0]	194 (19.3) [16.9–21.8]
Hypertension	179 (17.2) [15.0–19.7]	2 (6.3) [0.8–20.8]	177 (17.6) [15.3–20.1]
High cholesterol	139 (13.4) [11.4–15.6]	3 (9.4) [2.0–25.0]	136 (13.5) [11.5–15.8]
Osteoporosis/osteopenia/decreased bone density	120 (11.6) [9.7–13.7]	4 (12.5) [3.5–29.0]	116 (11.5) [9.6–13.7]
Thyroid problem	107 (10.3) [8.5–12.3]	2 (6.3) [0.8–20.8]	105 (10.4) [8.6–12.5]
Fractures	98 (9.4) [7.7–11.4]	4 (12.5) [3.5–29.0]	94 (9.3) [7.6–11.3]
Diabetes	90 (8.7) [7.0–10.5]	2 (6.3) [0.8–20.8]	88 (8.7) [7.1–10.7]
HIV/AIDS wasting syndrome	71 (6.8) [5.4–8.5]	2 (6.3) [0.8–20.8]	69 (6.9) [5.4–8.6]
Cardiac arrythmia	64 (6.2) [4.8–7.8]	5 (15.6) [5.3–32.8]	59 (5.9) [4.5–7.5]
Strokes	39 (3.8) [2.7–5.1]	3 (9.4) [2.0–25.0]	36 (3.6) [2.5–4.9]
Cognitive impairment	39 (3.8) [2.7–5.1]	2 (6.3) [0.8–20.8]	37 (3.7) [2.6–4.9]
**Mental health diagnoses currently living with**			
Depression	336 (32.3) [29.5–35.3]	8 (25.0) [11.5–43.4]	328 (32.6) [29.7–35.6]
Anxiety	306 (29.5) [26.7–32.3]	7 (21.9) [9.3–40.0]	299 (29.7) [26.9–32.6]
Sleep disorder	229 (22.0) [19.6–24.7]	2 (6.3) [0.8–20.8]	227 (22.5) [20.0–25.3]
Drug addiction	227 (21.9) [19.4–24.5]	10 (31.3) [16.1–50.0]	217 (21.5) [19.0–24.2]
Post-traumatic stress disorder	144 (13.9) [11.8–16.1]	2 (6.3) [0.8–20.8]	142 (14.1) [12.0–16.4]
Alcohol addiction	94 (9.1) [7.4–11.0]	2 (6.3) [0.8–20.8]	92 (9.1) [7.4–11.1]
Bipolar disorder	66 (6.4) [4.9–8.0]	0 (0) [0–10.9]	66 (6.6) [5.1–8.3]
Obsessive-compulsive disorder	34 (3.3) [2.3–4.5]	2 (6.3) [0.8–20.8]	32 (3.2) [2.2–4.5]
**Disabilities currently living with**			
Partial deafness	91 (8.8) [7.1–10.6]	4 (12.5) [3.5–29.0]	87 (8.6) [7.0–10.5]
Difficulty walking—require cane	81 (7.8) [6.2–9.6]	0 (0) [0–10.9]	81 (8.0) [6.4–9.9]

^1^ The following conditions were all reported by less than 5% in the overall cohort and in each categories, hence not included in the table: deep vein thrombosis/pulmonary embolism; renal problems; coronary artery diseases; inflammatory bowel disease; fibromyalgia; anorexia nervosa or bulimia nervosa; personality disorder; dementia; schizophrenia; complete deafness; partial blindness; difficulty walking—require wheelchair; speech difficulty; difficulty moving one or both arms. Note: BMI = body mass index. COPD = chronic obstructive pulmonary disease.

**Table 3 jpm-12-01294-t003:** Prevalence of comorbidities in participants from the Canadian HIV Women’s Sexual and Reproductive Health Cohort Study (CHIWOS), stratified by age and ethnicity.

Ethnicity	Indigenous			ACB			White/Other		
Age	16–39 *n* (%) [95% CI]	40–49 *n* (%) [95% CI]	50+ *n* (%) [95% CI]	16–39 *n* (%) [95% CI]	40–49 *n* (%) [95% CI]	50+ *n* (%) [95% CI]	16–39 *n* (%) [95% CI]	40–49 *n* (%) [95% CI]	50+ *n* (%) [95% CI]
Total (row%)	52 (30.1)	62 (35.8)	59 (34.1)	127 (33.9)	131 (34.9)	117 (31.2)	99 (20.2)	159 (32.3)	233 (47.5)
**Physical health diagnosis (lifetime prevalence)** ^1^									
Obesity (BMI > 30) ^2,3^	17 (32.7) [20.3–47.1]	19 (30.7) [19.6–43.7]	12 (20.3) [11.0–32.8]	28 (22.1) [15.2–30.3]	40 (30.5) [22.8–39.2]	50 (42.7) [33.6–52.2]	32 (32.3) [23.3–42.5]	42 (26.4) [19.7–34.0]	37 (15.9) [11.4–21.2]
Asthma/COPD ^2,3^	10 (19.2) [9.6–23.5]	23 (37.1) [25.2–50.3]	19 (32.2) [20.6–45.6]	8 (6.3) [2.8–12.0]	9 (6.9) [3.2–12.6]	9 (7.7) [3.6–14.1]	28 (28.3) [19.7–38.2]	55 (34.6) [27.2–42.5]	81 (34.8) [28.7–41.3]
Arthritis, osteoarthritis ^2,3^	6 (11.5) [4.4–23.4]	27 (43.6) [31.0–56.7]	28 (47.5) [34.3–60.9]	<5%	9 (6.9) [3.2–12.6]	20 (17.1) [10.8–25.2]	8 (8.1) [3.6–15.3]	33 (20.8) [14.7–27.9]	84 (36.1) [29.9–42.6]
Cancer or pre-cancer ^2^	9 (17.3) [8.2–30.3]	14 (22.6) [12.9–35.0]	12 (20.3) [11.0–32.8]	11 (8.7) [4.4–15.0]	13 (9.9) [5.4–16.4]	18 (15.4) [9.4–23.2]	25 (25.3) [17.1–35.0]	41 (25.8) [19.2–33.3]	64 (27.5) [21.8–33.7]
Chronic pains for other causes than arthristis/osteoarthritis requiring medication ^2,3^	7 (13.5) [5.6–25.8]	24 (38.7) [26.6–51.9]	19 (32.2) [20.6–45.6]	<5%	9 (6.9) [3.2–12.6]	13 (11.1) [6.1–18.3]	10 (10.1) [5.0–17.8]	44 (27.7) [20.9–35.3]	67 (28.8) [23.0–35.0]
Hypertension ^3^	5 (9.6) [3.2–21.0]	9 (14.5) [6.9–25.8]	15 (25.4) [15.0–38.4]	9 (7.1) [3.3–13.0]	24 (18.3) [12.1–26.0]	41 (35.0) [26.5–44.4]	<5%	19 (12.0) [7.4–18.0]	53 (22.8) [17.5–28.7]
High cholesterol ^2,3^	3 (5.8) [1.2–15.9]	4 (6.5) [1.8–15.7]	9 (15.3) [7.2–27.0]	<5%	13 (9.9) [5.4–16.4]	15 (12.8) [7.4–20.3]	10 (10.1) [5.0–17.8]	21 (13.2) [8.4–19.5]	60 (25.8) [20.3–31.9]
Osteoporosis/osteopenia/decreased bone density ^2,3^	4 (7.7) [2.1–18.5]	8 (12.9) [5.7–23.9]	17 (28.8) [17.8–42.1]	<5%	<5%	13 (11.1) [6.1–18.3]	5 (5.1) [1.7–35.0]	12 (7.6) [4.0–12.8]	55 (23.6) [18.3–29.6]
Thyroid problem	5 (9.6) [3.2–21.0]	8 (12.9) [5.7–23.9]	6 (10.2) [3.8–20.8]	<5%	<5%	10 (8.6) [4.2–15.2]	7 (7.1) [2.9–14.0]	25 (15.7) [10.4–22.3]	38 (16.3) [11.8–21.7]
Fractures	7 (13.5) [5.6–25.8]	8 (12.9) [5.7–23.9]	13 (22.0) [12.3–34.7]	<5%	<5%	6 (5.1) [1.9–10.8]	8 (8.1) [3.6–15.3]	17 (10.7) [6.4–16.7]	35 (15.0) [10.7–20.3]
Diabetes	<5%	<5%	7 (11.9) [4.9–22.9]	<5%	12 (9.2) [4.8–15.5]	20 (17.1) [10.8–25.2]	5 (5.1) [1.7–35.0]	15 (9.4) [5.4–15.1]	25 (10.7) [7.1–15.4]
HIV/AIDS wasting syndrome	<5%	<5%	<5%	<5%	11 (8.4) [4.3–14.5]	9 (7.7) [3.6–14.1]	6 (6.1) [2.3–12.7]	10 (6.3) [3.1–11.3]	26 (11.2) [7.4–15.9]
Cardiac arrythmia	<5%	7 (11.3) [4.7-21.9]	6 (10.2) [3.8–20.8]	<5%	<5%	6 (5.1) [1.9–10.8]	5 (5.1) [1.7–35.0]	11 (6.9) [3.5–12.0]	20 (8.6) [5.3–12.9]
Strokes	<5%	4 (6.5) [1.8–15.7]	6 (10.2) [3.8–20.8]	0	<5%	6 (5.1) [1.9–10.8]	<5%	<5%	13 (5.6) [3.0–9.4]
**Mental health diagnoses currently living with** ^1^									
Depression ^2^	23 (44.2) [30.5–58.7]	32 (51.6) [38.6–64.5]	27 (45.8) [32.7–59.2]	18 (14.2) [8.6–21.5]	20 (15.3) [9.6–22.6]	27 (23.1) [15.8–31.8]	31 (31.3) [22.4–41.4]	70 (44.0) [36.2–52.1]	88 (37.8) [31.5–44.3]
Anxiety ^2^	19 (36.5) [23.6–51.0]	33 (52.2) [40.1–66.0]	24 (40.7) [18.1–54.3]	12 (9.5) [5.0–15.9]	11 (8.4) [4.3–14.5]	16 (13.7) [8.0–21.3]	40 (40.4) [30.7–50.7]	71 (44.7) [36.8–52.7]	80 (34.3) [28.3–40.8]
Sleep disorder ^2^	10 (19.2) [9.6–23.5]	18 (29.0) [18.2–41.9]	15 (25.4) [15.0–38.4]	11 (8.7) [4.4–15.0]	12 (9.2) [4.8–15.5]	18 (15.4) [9.4–23.2]	23 (23.2) [15.3–32.8]	47 (29.6) [22.6–37.3]	75 (32.2) [26.2–38.6]
Drug addiction ^2,3^	35 (67.3) [52.9–79.7]	39 (62.9) [49.7–74.8]	27 (45.8) [32.7–59.2]	<5%	<5%	<5%	35 (35.4) [26.0–45.6]	45 (28.3) [21.5–36.0]	41 (17.6) [12.9–23.1]
Post-traumatic stress disorder ^2^	15 (28.9) [17.1–43.1]	22 (35.5) [23.7–48.7]	28 (30.5) [34.3–60.9]	<5%	<5%	6 (5.1) [1.9–10.8]	14 (14.1) [8.0–22.6]	35 (12.5) [15.8–29.3]	29 (12.5) [8.5–17.4]
Alcohol addiction ^2^	11 (21.2) [11.1–34.7]	23 (37.1) [25.2–50.3]	17 (28.8) [17.8–42.1]	<5%	<5%	<5%	13 (13.1) [7.2–21.4]	13 (8.2) [4.4–13.6]	13 (5.6) [3.0–9.4]
Bipolar disorder	7 (13.5) [5.6–25.8]	9 (14.5) [6.9–25.8]	6 (10.2) [3.8–20.8]	<5%	0	<5%	11 (11.1) [5.7–19.0]	16 (10.1) [5.9–15.8]	13 (5.6) [3.0–9.4]
Anorexia nervosa or bulimia nervosa	4 (7.7) [2.1–18.5]	<5%	<5%	0	0	0	<5%	<5%	<5%
Personality disorder	4 (7.7) [2.1–18.5]	8 (12.9) [5.7–23.9]	<5%	<5%	<5%	0	8 (8.1) [3.6–15.3]	11 (6.9) [3.5–12.0]	<5%
Obsessive-compulsive disorder	5 (9.6) [3.2–21.0]	<5%	5 (8.5) [2.8–18.7]	<5%	0	0	<5%	9 (5.7) [2.6–10.5]	<5%
**Disabilities currently living with** ^1^									
Partial deafness	4 (7.7) [2.1–18.5]	10 (16.1) [8.0–27.7]	16 (27.1) [16.4–40.3]	<5%	<5%	<5%	<5%	13 (8.2) [4.4–13.6]	39 (16.7) [12.2–22.2]
Partial blindness	6 (11.5) [4.4–23.4]	5 (9.7) [2.7–17.8]	6 (10.2) [3.8–20.8]	<5%	<5%	<5%	<5%	11 (6.9) [3.5–12.0]	<5%
Difficulty walking—require cane	3 (5.8) [1.2–15.9]	6 (9.7) [3.6–19.9]	9 (15.3) [7.2–27.0]	<5%	<5%	8 (6.8) [3.0–13.0]	<5%	11 (6.9) [3.5–12.0]	37 (15.9) [11.4–21.2]
Difficulty walking—require wheelchair	0	<5%	4 (6.8) [1.9–16.5]	0	0	<5%	0	<5%	<5%
Difficulty moving one or both arms	<5%	<5%	5 (8.5) [2.8–18.7]	<5%	<5%	<5%	<5%	<5%	<5%

^1^ The following conditions were all reported by less than 5% in all categories, hence not included in the table: deep vein thrombosis/pulmonary embolism; fibromyalgia; dementia; schizophrenia; complete deafness; speech difficulty. ^2^ Indicates statistically significant difference across ethnic groups of same age. ^3^ Indicates statistically significant difference across age groups of a same ethnicity. Note: ACB = African, Caribbean, Black; BMI = body mass index. COPD = chronic obstructive pulmonary disease.

## Data Availability

Data are available upon request to corresponding author. More information on the CHIWOS study and its policies can also be found on www.chiwos.ca.

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
