# Peer review of "Prevalence of Physical Health, Mental Health, and Disability Comorbidities among Women Living with HIV in Canada"

_jpm, 2022, doi:10.3390/jpm12081294_

Round 1
Reviewer 1 Report
I don’t have any comment on the manuscript as the manuscript is well written and informative. However, I have a suggestion for font color of the data represented in the table as the gray scale is not visible. Please change the font color.
Correct the minor mistakes in the figure 2 caption.
Author Response
I don’t have any comment on the manuscript as the manuscript is well written and informative. However, I have a suggestion for font color of the data represented in the table as the gray scale is not visible. Please change the font color. Correct the minor mistakes in the figure 2 caption.
Response: Thank you for this positive feedback. We changed the font color in the tables; everything is now black. We also corrected the figure 2 caption.
Reviewer 2 Report
Review, Heer et al., Journal of Personalized Medicine, 2022: Prevalence of physical health, mental health and disability comorbidities among women living with HIV in Canada
Summary
In the study by Heer et al, the group proposed to measure the prevalence of comorbidities among women living with HIV in Canada.
They reported that among 1 039 participants, the most prevalent comorbidities were depression (32.3%), anxiety (29.5%), obesity (26.7%), asthma/COPD (23.3%), sleep disorder (22%), drug addiction (21.9%) and arthritis/osteoarthritis (20.9%). These results highlight the complexity of HIV care and the importance of personalized health care.
Overall, this manuscript is well written and analysis well conducted. The 2000s represent a high point for ART, many drugs are now available to allow personalized medicine to fight the virus, but non-AIDS diseases are often linked in the background. Your work has shown the importance of reinforcing the need to treat them as well in all the populations. I just have some minor comments before publication.
Minor comments
Introduction
1. Line 56 you should mention the chronic inflammation and immune activation.
2. Paragraph line 62, you should introduce the term of inflammageing.
3. Paragraph that starts line 68 is not well constructed. Authors should begin by the representation of women living with HIV worldwide then focus in the Canada. The part that starts line 75 should be at the beginning to explain why women are less represented in HIV studies.
4. Lines 71, 72, women were compared with who? Uninfected women, men, …?
Materials and Methods
1. Authors should create a supplemental files with a schematic representation for the questionnaire and reduce this part which is hard to read.
2. Lines 164-168 should be at the beginning of this part.
Results
1. Can you provide the acronym of ACB and PTSD. Be sure that all your acronyms are described.
2. Transgender women information (baseline characteristics such as in table 1) should be included in the 3.1 part and Table 1.
3. Table 3, highlight p-value significant in bold because it is not clear here.
4. Authors observed some association following age and ethnicity with medical disorder but did you try to correlate all medical information together to see if a particular profile represent, for example, people with anxiety.
Discussion
1. Line 412, a space is missing between professionals and which.
Author Response
We thank the reviewer for these helpful comments. Here are our specific responses to the minor comments.
Introduction
- Line 56 you should mention the chronic inflammation and immune activation.
Response 1: The line was modified accordingly, and an additional reference was added to support the statement.
- Paragraph line 62, you should introduce the term of inflammageing.
Response 2: The term inflammaging was introduced, and additional reference was added to better support the sentence.
- Paragraph that starts line 68 is not well constructed. Authors should begin by the representation of women living with HIV worldwide then focus in the Canada. The part that starts line 75 should be at the beginning to explain why women are less represented in HIV studies.
Response 3: The paragraph was modified as per suggested.
- Lines 71, 72, women were compared with who? Uninfected women, men, …?
Response 4: We added “as compared with women not living with HIV”.
Materials and Methods
- Authors should create a supplemental files with a schematic representation for the questionnaire and reduce this part which is hard to read.
Response 1: Although we really considered this suggestion, we feel this information is very important for the reader to understand how the main outcomes of interest were measured. We would prefer to keep this section in the main manuscript.
- Lines 164-168 should be at the beginning of this part.
Response 2: We moved these lines to earlier in the section.
Results
- Can you provide the acronym of ACB and PTSD. Be sure that all your acronyms are described.
Response 1: ACB was already defined in line 156, but it was indeed missing from the table foot notes, which we corrected. PTSD was also already defined in line 146.
- Transgender women information (baseline characteristics such as in table 1) should be included in the 3.1 part and Table 1.
Response 2: We added the characteristics of transgender participants in section 3.1, but not in Table 1 to avoid making this table too heavy. Note that while doing this, we noticed that the median age of all participants had not been updated correctly from our last analyses and we corrected that too.
- Table 3, highlight p-value significant in bold because it is not clear here.
Response 3: Two proportions are considered statistically significantly different when their 95%CI do not overlap. Since comparisons can be made across ages within a same ethnicity or across ethnicities for same age groups, the bolding would probably add confusion as to which comparison it refers too. We would prefer to keep the table as is.
- Authors observed some association following age and ethnicity with medical disorder but did you try to correlate all medical information together to see if a particular profile represent, for example, people with anxiety.
Response 4: This is a very interesting suggestion but would probably best be done through latent class analysis, and would best be presented as separate manuscript.
Discussion
- Line 412, a space is missing between professionals and which.
Response 1: This mistake was corrected.